# Three phylogenetic groups have driven the recent population expansion of *Cryptococcus neoformans*

P.M. Ashton [1,2], L.T. Thanh[1], P.H. Trieu[1], D. Van Anh[1], N.M. Trinh[1], J. Beardsley[1,2,3], F. Kibengo [4], W. Chierakul [5], D.A.B. Dance [2,6,7], S. Rattanavong[6], V. Davong[6], L.Q. Hung[8], N.V.V. Chau[9], N.L.N. Tung[9], A.K. Chan[10,11], G.E. Thwaites[1,2], D.G. Lalloo [12], C. Anscombe[1,2], L.T.H. Nhat[1], J. Perfect[13], G. Dougan[14,15,16], S. Baker[1,2,14,16], S. Harris[15] & J.N. Day [1,2]

*Cryptococcus neoformans* (*C. neoformans* var. *grubii*) is an environmentally acquired pathogen causing 181,000 HIV-associated deaths each year. We sequenced 699 isolates, primarily *C. neoformans* from HIV-infected patients, from 5 countries in Asia and Africa. The phylogeny of *C. neoformans* reveals a recent exponential population expansion, consistent with the increase in the number of susceptible hosts. In our study population, this expansion has been driven by three sub-clades of the *C. neoformans* VNIa lineage; VNIa-4, VNIa-5 and VNIa-93. These three sub-clades account for 91% of clinical isolates sequenced in our study. Combining the genome data with clinical information, we find that the VNIa-93 sub-clade, the most common sub-clade in Uganda and Malawi, was associated with better outcomes than VNIa-4 and VNIa-5, which predominate in Southeast Asia. This study lays the foundation for further work investigating the dominance of VNIa-4, VNIa-5 and VNIa-93 and the association between lineage and clinical phenotype.

[1] Wellcome Trust Asia Programme, Oxford University Clinical Research Unit, 764 Vo Van Kiet, Ho Chi Minh City, Vietnam. [2] Nuffield Department of Medicine, Centre for Tropical Medicine and Global Health, University of Oxford, Oxford OX3 7FZ, UK. [3] Marie Bashir Institute, University of Sydney, Sydney 2050 NSW, Australia. [4] MRC/UVRI and LSHTM Uganda Research Unit, Entebbe, Uganda. [5] Mahidol Oxford Tropical Medicine Research Unit, Bangkok, Thailand. [6] Lao–Oxford–Mahosot Hospital–Wellcome Trust Research Unit, Vientiane, Laos. [7] Faculty of Infectious and Tropical Diseases, London School of Hygiene and Tropical Medicine, London WC1E 7HT, UK. [8] Cho Ray Hospital, Ho Chi Minh City, Vietnam. [9] Hospital for Tropical Diseases, Ho Chi Minh City, Vietnam. [10] Sunnybrook Health Sciences Centre, University of Toronto, Toronto M4N 3M5 ON, Canada. [11] Dignitas International, Zomba, Malawi. [12] Liverpool School of Tropical Medicine, Liverpool L3 5QA, UK. [13] Department of Medicine and Department of Molecular Genetics and Microbiology, Division of Infectious Diseases, Duke University, Durham, NC 27710, USA. [14] Wellcome Trust-Cambridge Centre for Global Health Research, Cambridge CB2 0XY, UK. [15] Pathogen Genomics, The Wellcome Trust Sanger Institute, Wellcome Trust Genome Campus, Cambridge CB10 1SA Cambridgeshire, UK. [16] Department of Medicine, University of Cambridge, Cambridge CB2 0SP, UK. Correspondence and requests for materials should be addressed to J.N.D. (email: jday@oucru.org)

Cryptococcus neoformans is an opportunistic fungal pathogen that primarily affects people with cell-mediated immune defects, particularly those living with HIV. There are an estimated 223,100 incident cases of cryptococcal meningitis per year in HIV patients with CD4 counts of <100 cells/μl, resulting in 181,100 deaths[1]. Cryptococcus neoformans var. grubii (hereafter C. neoformans), one of two varieties of C. neoformans, accounts for the vast majority of cryptococcal meningitis cases globally, particularly in the tropical and sub-tropical regions, which bear the heaviest disease burden[1,2].

The population structure of C. neoformans consists of at least three lineages, VNI, VNII and VNB. Two of these, the frequently isolated VNI and the rarely observed VNII, are clonal and globally distributed[3–5], while VNB is very diverse but rarely isolated outside sub-Saharan Africa[3] and South America[6]. Sequencing of strains from patients with relapsed disease has indicated that microevolution occurs during infection, with typically 0–6 single-nucleotide polymorphisms (SNPs) occurring over a median relapse period of 146 days[7]. Other studies have described a broad view of the three main molecular types, VNI, VNII and VNB, analysing 150–400 total isolates, and placing clinical isolates into the context of environmental strains[8–10]. Within VNI, three distinct, but still recombining, sub-lineages have been identified, two of which (VNIa and VNIb) are globally distributed, while VNIc is limited to southern Africa. Genomic data has revealed that VNI and VNII have more recent migrations than VNB, with nearly clonal isolates found in disparate geographic regions[9], although this has not yet been investigated on a fine scale.

So far, our understanding of the population structure of C. neoformans in the Asia and Pacific region, the second highest prevalence region after sub-Saharan Africa[1], has been based upon low resolution methods such as multi-locus sequence typing (MLST) and amplified fragment length polymorphism (AFLP)[4,11–16]. These data show that C. neoformans in Southeast Asia is highly clonal, with considerable gene flow between countries within the region, and less connectivity with other continents[4]. Recently, the first study focussing on whole-genome data from the region has been reported, which identified 165 kbp of sequence specific to ST5[15], a sequence type seen more frequently in HIV uninfected patients, the majority of whom have no identified underlying immune-suppression[11,15]. The predilection of ST5 to infect HIV-uninfected patients is not the only reported association between a C. neoformans lineage and a clinical phenotype. Infections with VNB[17] and VNI ST93[18] have been reported to have worse outcomes in HIV-infected patients in Southern Africa and Eastern Africa, respectively.

Previously, we have undertaken several prospective, descriptive and randomized controlled intervention trials in Southeast Asia and East/Southeast Africa[19–21]. Here, we use whole-genome sequence analysis of 699 Cryptococcus isolates to describe the population structure of C. neoformans causing disease in these populations in high resolution, and combine this information with metadata from these trials to relate this to disease phenotype. We perform a detailed analysis of the obtained phylogenies, phylogeographic analysis, recombination analysis, phylo-temporal analysis, compare the mitochondrial and chromosome phylogenies, and re-analyse outcomes from previous clinical trials in the context of this new genomic information.

## Results

**Isolate characteristics.** We sequenced 699 isolates, including all available clinical isolates and 3 environmental C. neoformans isolates. They were from Vietnam (n = 441), Laos (n = 73), Thailand (n = 40), Uganda (n = 132) and Malawi (n = 13). There were 696 clinical isolates from 695 patients, and 3 environmental isolates (all from Vietnam). Of the total, 682 were C. neoformans, 12 were C. gattii and 5 (all from Uganda) were putative hybrids between C. neoformans and C. deneoformans. All environmental isolates were C. neoformans. There were 618 isolates from HIV-infected patients and 78 from HIV-uninfected patients. Of the 682 C. neoformans, there were 681 isolates with mating type alpha and 1 isolate from Vietnam with mating type a.

**Whole-genome sequencing of VNI.** Six hundred and seventy eight (99.4%) of our C. neoformans isolates were VNI, while four were VNII (Supplementary Fig. 1, Supplementary Data 1). To provide context for our isolates, all 185 VNI genomes sequenced by Desjardins et al.[8] (160 clinical, 25 environmental, full details available in Supplementary Data 1) were included in subsequent phylogenetic analyses. We ensured technical comparability of our methods of phylogenetic analysis with those of Desjardins et al.[8] by comparing our results for the Desjardins data with their reported results (Supplementary Fig. 2).

A phylogenetic tree (Fig. 1) was derived from the 325,812 variant positions in the core genome of the 863 C. neoformans VNI. Of the novel C. neoformans isolates presented here, 668 were VNIa (97.9%), 10 were VNIb (1.5%); none were VNIc. Figure 1 shows that the population structure of VNIa is dominated by three common and highly clonal sub-clades, while VNIb and VNIc are more heterogenous. VNIa, VNIb and VNIc isolates were isolated from 14, 10 and 2 countries on 5, 6 and 1 continent(s), respectively (Supplementary Tables 1 and 2). VNIa was predominant, accounting for 548 of 549 (99.8%) isolates in Asia and 163 of 274 (59.5%) strains in Africa. When isolates from Botswana, an established outlier in terms of Cryptococcus neoformans diversity, were excluded, the proportion of VNIa isolates in Africa was 84.3% (134 out of 159) of all VNI isolates. The H99 reference genome belonged to VNIb.

Nine distinct clusters were identified using PCA and K-means clustering (Supplementary Fig. 3). We extended the naming scheme of Desjardins et al.[8] to refer to the sub-clades within VNIa as VNIa-4, VNIa-5, VNIa-93 and VNIa-32 after the predominant MLST sequence type in each clade. Two clusters contained only isolates with novel sequence types (STs), which we refer to as VNIa-X and VNIa-Y. The previously described VNIb and VNIc lineages were also identified as distinct clusters. We grouped the remaining polyphyletic VNI isolates, which did not fall into any PCA cluster into VNI-outlier. The numbers of each phylogenetic group isolated from HIV positive patients from each country are presented in Table 1.

While each country with more than 30 VNI isolates from HIV infected individuals had a dominant or, in the case of Vietnam, co-dominant sub-clade(s), there were minority sub-clades present in every country analysed (Supplementary Fig. 4). For example, VNIa-93, the dominant lineage in Uganda, was also present in Vietnam (12%). Similarly, Uganda and Botswana had low prevalence of typically Southeast Asian sub-clades, such as VNIa-4 (Uganda = 1.6%, Botswana = 2.9%) and VNIa-5 (Uganda = 6.5%, Botswana = 4.9%).

**Phylogenetic analysis of sub-clades within VNIa.** We performed fine-scale genomic epidemiological analyses of VNIa for every sub-clade with at least 50 isolates from this study, that is, VNIa-4, VNIa-5 and VNIa-93. These sub-clades accounted for 89% of the isolates sequenced as part of our study, with VNIa-4 accounting for 41%, VNIa-5 for 29% and VNIa-93 for 20%. To maximize the phylogenetic resolution within these sub-clades, within sub-clade reference genomes were generated using PacBio sequencing

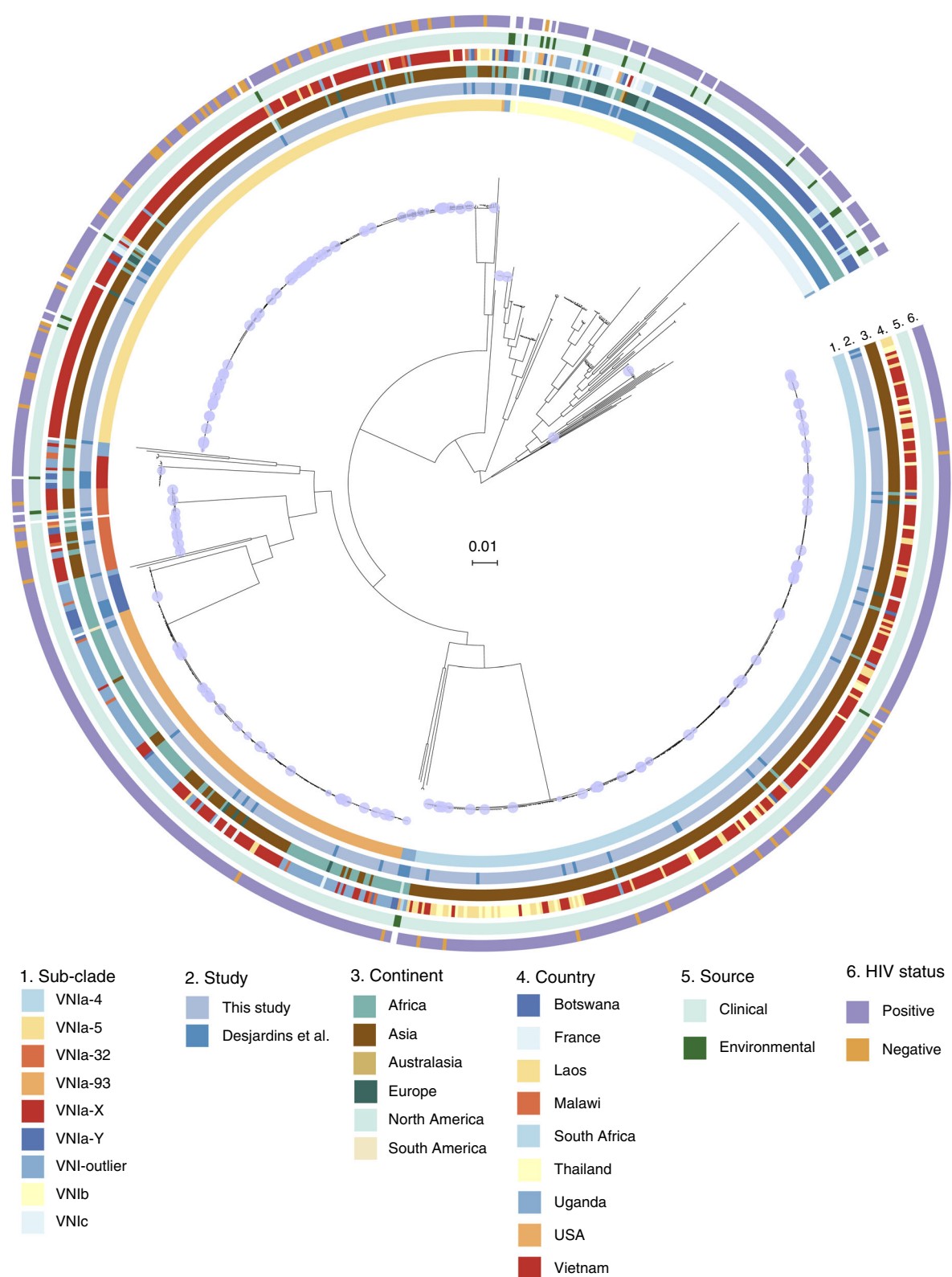

**Fig. 1** A whole-genome single-nucleotide polymorphism (SNP) phylogeny of all VNI in this study and Desjardins et al.[8], constructed from 325,812 variable positions. Nodes with <80% bootstrap support are highlighted by a blue circle. Scale bar is genetic distance in number of substitutions per site

(PacBio genome assemblies available via FigShare https://doi.org/10.6084/m9.figshare.6060686). The median SNP distance of the VNIa-4, VNIa-5 and VNIa-93 strains to the within sub-clade reference genome was 277 (standard deviation (SD) = 142), 338 (SD = 236) and 361 (SD = 44) SNPs, respectively, compared with

47,619 (SD = 196), 46,218 (SD = 245) and 48,763 (SD = 262) to the H99 reference genome.

**Recombination within sub-clades.** Before deriving per sub-clade phylogenies from which genomic-epidemiological characteristics

**Table 1 Frequency of isolation of VNI sub-clades from HIV-infected patients in each country from both this study and Desjardins et al.[8]**

| Country | VNIa-4 | VNIa-5 | VNIa-93 | VNIc | VNIb | VNIa-32 | VNIa-Y | VNIa-X | VNIa-outlier | Total |
|---|---|---|---|---|---|---|---|---|---|---|
| Vietnam | 175 | 129 | 44 | | 1 | 15 | | 1 | 1 | 366 |
| Uganda | 2 | 8 | 84 | | 10 | 3 | 7 | 3 | 5 | 122 |
| Botswana | 3 | 5 | 3 | 74 | 2 | 3 | 6 | 3 | 3 | 102 |
| Laos | 57 | 6 | 2 | | | | | | | 65 |
| Thailand | 38 | 4 | | | | | | | | 42 |
| France | 2 | 4 | 4 | | 15 | | | | | 25 |
| S. Africa | 1 | 1 | | 6 | 6 | | | 2 | 1 | 17 |
| Malawi | | 3 | 5 | | | 2 | 1 | 2 | | 13 |
| Togo | | | | | 2 | | | | | 2 |
| India | | | | | | 1 | | | | 1 |
| Brazil | | | 1 | | | | | | | 1 |
| Argentina | | | | | 1 | | | | | 1 |
| Australia | | | | | 1 | | | | | 1 |
| USA | | 1 | | | | | | | | 1 |
| China | | 1 | | | | | | | | 1 |
| Japan | | 1 | | | | | | | | 1 |
| Tanzania | | | | | | | | | 1 | 1 |
| Total | 278 | 163 | 143 | 80 | 38 | 24 | 14 | 11 | 11 | 762 |

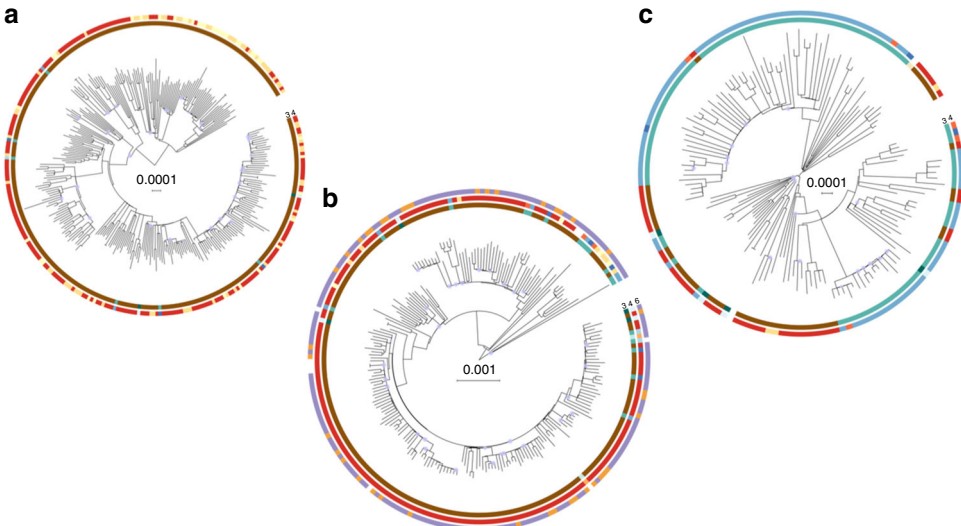

**Fig. 2** Within sub-clade phylogenetic trees for **a** VNIa-4, **b** VNIa-5 and **c** VNIa-93. Rings are numbered and coloured according to Fig. 1. Nodes with <80% bootstrap support are denoted with blue circles. Trees were constructed from 24,956 (VNIa-4), 22,894 (VNIa-5) and 11,056 (VNIa-93) variable positions. Scale bar is genetic distance in number of substitutions per site

can be inferred, we quantified the extent to which recombination plays a role in generation of diversity within sub-clades. Recombination within sub-clades was investigated by assessing the degree of linkage disequilibrium (LD). LD was assessed for all within sub-clade SNPs with a minor allele frequency of 0.1 or greater. There was limited decay of LD, indicating minimal ongoing recombination (Supplementary Fig. 5).

**Sub-clades include isolates from disparate locations**. One of the most striking patterns observed in the per-sub-clade phylogenies is the interspersion of isolates from different countries and different continents throughout the phylogeny (see Fig. 2a–c), indicating frequent international and intercontinental transmissions. We used parsimony analysis to quantify the minimum number of international transmission events that explain the current geographic distribution of strains. VNIa-4 had the largest number of international transmission events (primarily between Thailand, Laos and Vietnam) as a proportion of total internal branches (95% confidence interval (CI) in parentheses, VNIa-4 = 13% (11–16%), VNIa-5 = 8% (6–11%), VNIa-93 = 10% (7–14%)), while VNIa-93 had the highest proportion of intercontinental branches (VNIa-4 = 1% (0–2%), VNIa-5 = 5% (3–7%), VNIa-93 = 7% (5–10%)).

**Notable within sub-clade phylogenetic features**. A striking feature of the within sub-clade phylogenies is the combination of long terminal branch lengths and short internal branches. The median numbers of SNPs represented by the internal branch lengths compared with the terminal branch lengths are 4.5 vs. 60 for VNIa-4 (P value from Kolmogorov–Smirnov test = $7 \times 10^{-70}$), 3 vs. 77.5 for VNIa-5 (P value = $1 \times 10^{-53}$) and 6 vs. 44.5 for VNIa-93 (P value = $4 \times 10^{-19}$) (Supplementary Fig. 6).

There were a total of 18,071, 17,593 and 7163 terminal branch SNPs in VNIa-4, VNIa-5 and VNIa-93, respectively. We had only

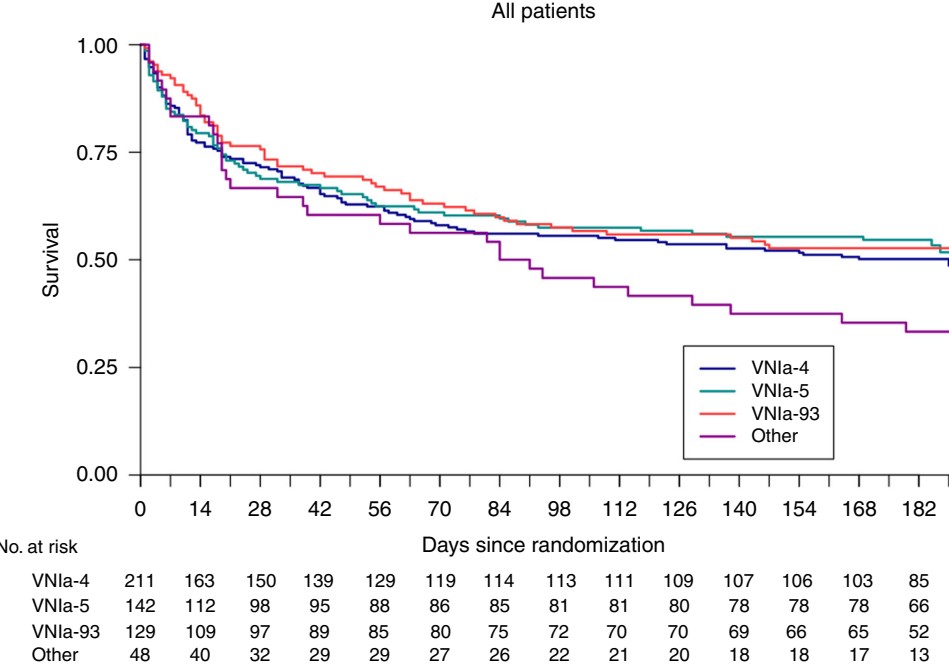

**Fig. 3** Kaplan–Meier survival estimates up to 6 months for all 530-HIV infected patients enroled in one of two clinical trials (Day et al.[20]; Beardsley et al.[21]) with whole-genome sequencing results for their infecting isolate

five environmental strains in our dataset (one VNIa-4 and four VNIa-5, three from our study and two from Desjardins et al.[8]), and they had a similar mean terminal branch length (75 SNPs). There were 263, 294 and 31 variants (1.5%, 1.8% and 0.4% of total), which occurred more than once on different terminal branches in VNIa-4, VNIa-5 and VNIa-93. However, most of these (VNIa-4, 52%; VNIa-5, 60%; and VNIa-93, 65%) were in intergenic regions (i.e. not in coding sequence, 3′- or 5′- untranslated region or introns). We manually investigated any gene containing a variant that occurred as a homoplasy in three or more strains for recognized links with virulence or host interactions, but had no hits. The average d$N$/d$S$ of SNPs in the terminal branches were 0.84, 0.82 and 0.84 in VNIa-4, VNIa-5 and VNIa-93, respectively.

Another striking feature of the within sub-clade trees was the number of polytomies. All internal branches that represented 0 SNPs were collapsed, resulting in 78, 65 and 35 collapsed branches in 46, 36 and 21 distinct polytomies (defined as nodes with more than 2 children, after branches of 0 SNPs were collapsed) in VNIa-4, VNIa-5 and VNIa-93, respectively. The collapsed branches as a proportion of the total number of branches in each sub-clade were 13%, 15% and 12% in VNIa-4, VNIa-5 and VNIa-93, respectively. The median number of branches resulting from a polytomy event was 3 in all sub-clades, while the maximum was 9, 11 and 6 in VNIa-4, VNIa-5 and VNIa-93, respectively (Supplementary Table 3). For VNIa-4, 14 of 29 (48%) polytomies were international (i.e. strains in the polytomy were isolated from more than one country) and 1 (3%) of these was intercontinental. For VNIa-5, 10 of 24 (42%) polytomies were international and 6 (25%) were intercontinental. For VNIa-93, 4 of 21 (19%) polytomies were international and 1 (5%) of these was intercontinental. The maximum time separating the sampling date of two isolates descending directly from the same polytomy (i.e. not separated via an internal branch representing >0 SNPs) was 10 years for VNIa-4, 15 years for VNIa-5 and 8 years for VNIa-93. The median time range spanned by polytomies was 5.5, 5 and 1 year(s) for VNIa-4, VNIa-5 and

VNIa-93, respectively. Genome sequences from isolates from both our study and that of Desjardins et al.[8] belonged to the same polytomies.

We investigated the presence of nonsense mutations in DNA mismatch repair genes. We found six isolates with nonsense mutations in one of the 34 DNA mismatch repair genes we investigated; all six isolates had different mutations, although three strains had mutations in the same gene (CNAG_02073) (Supplementary Table 6). The terminal branch lengths of isolates with nonsense mutations in DNA mismatch repair genes were not significantly longer than isolates without these mutations. The mean terminal branch length of isolates with nonsense mutations was 0.006 (SD = 0.012), while mean terminal branch length of isolates without these mutations was 0.002 (SD = 0.009). There was no significant difference between the distributions of branch lengths (Wilcoxon's rank-sum statistic, $P$ value = 0.15).

**Within sub-clade temporal patterns**. The majority of isolates in our study were collected during two clinical trials, which recruited patients between 2004–2010 and 2013–2015 (Supplementary Fig. 7A). As the first clinical trial only recruited patients in Vietnam, this is the only country for which we have considerable temporal range. These data show that two sub-clades, VNIa-4 and VNIa-5, have been predominant in every year in which more than five samples were taken since 2004 (Supplementary Fig. 7B). The prevalence of VNIa-32 appears to have declined; in 2004 it accounted for 12% (4/34) of *C. neoformans* collected, while there were no cases of this sub-clade observed in 2014 (0/40), the last year of collection.

We found a lack of clock like evolution within all three sub-clades. The slope of the trend-line between time of isolation and root to tip distance was negative for both VNIa-4 and VNIa-5. There was a poor correlation between time of isolation and distance from the root in the tree for all three sub-clades (correlation co-efficient −0.07, −0.22 and 0.32 for VNIa-4, VNIa-5 and VNIa-93) (Supplementary Fig. 8).

**Association between sub-clade and clinical outcome.** We used data from our recent randomized controlled trials of treatment for HIV-associated cryptococcal meningitis patients to define the effect of sub-clade on survival until 10 weeks or 6 months after randomization. We used a Cox proportional hazards regression model with sub-clade as the main covariate, adjusted for country and treatment. Complete data were available from 530 patients. The survival over 6 months is illustrated in Fig. 3. Infections with VNIa-93 were associated with a significantly reduced risk of death by both 10 weeks and 6 months (hazard ratios (HR) 0.45, 95% CI 0.26–0.76, $P = 0.003$ and 0.60, 95% CI 0.39-0.94, $P = 0.024$, respectively) compared with lineage VNIa-4 infections. There were no differences in outcomes between infections with VNIa-4 and any other lineage (see Supplementary Tables 4 and 5).

**Association between VNIa-5 and HIV-uninfected patients.** Vietnam was the only country with more than 10 isolates of *C. neoformans* from HIV-uninfected people. Therefore, only isolates from Vietnam were included in this analysis. Thirty five percent of HIV-infected patients were infected with VNIa-5, compared with 75% of HIV-uninfected patients (Fisher's exact test, odds ratio 5.4, 95% CI 2.8–10.8, $P < 10^{-8}$). Isolates from HIV-uninfected patients are interspersed throughout the entire VNIa-5 phylogeny, implying that all strains of this cluster may have the potential to cause infection in such hosts. HIV infection status had no significant association with the terminal branch length of VNIa-5 isolates.

**VNIa-5 defining SNPs.** Due to the association between VNIa-5 and disease in HIV-uninfected patients, we were interested in SNPs which define VNIa-5. Ancestral sequence reconstruction identified 7465 SNPs between the 'origin' of VNIa-5 and the most recent common ancestor (MRCA) of VNIa-5, which were 95% sensitive and specific for VNIa-5. There were 1868 non-synonymous SNPs, distributed among 1220 genes. The d$N$/d$S$ ratio was calculated for all genes with SNPs on the VNIa-5 defining branch; there were no genes known to be associated with virulence or interaction with the host that had extremes of d$N$/d$S$ ratio. The overall d$N$/d$S$ ratio of genic SNPs on this branch was 0.33, compared with the SNPs on the VNIa-4 defining branch, which had an overall d$N$/d$S$ of 0.38. There were seven genes with nonsense SNPs, introducing premature stop codons into five hypothetical proteins, one E3 ubiquitin-protein ligase (CNAG_04262) and a metacaspase, a cysteine protease involved in cell apoptosis (CNAG_06787).

**Mitochondrial sequence.** A maximum likelihood phylogeny was derived for the SNPs identified in the mitochondrial DNA (mtSNP) of *C. neoformans* VNI (Supplementary Fig. 9B). When the mtSNP tree was compared with the whole genome SNP (wgSNP) tree (Supplementary Fig. 9B), some sub-clades were phylogenetically congruous, while others were not. VNIa-4, VNIa-5, VNIa-32 and VNIa-Y were all monophyletic within the mtSNP tree, in agreement with the wgSNP tree (Supplementary Fig. 9A). For VNIa-93, 144 out of 145 isolates were paraphyletic, with the monophyletic VNIa-32 and VNIa-Y nested within the VNIa-93 genotype, while VNIa-X was identical to the majority mtSNP genotype of VNIa-93. In the mitochondrial phylogeny VNIb is paraphyletic, giving rise to two sub-clades of VNIc, the first contained 19 isolates, while the second is a singleton, and two VNI-outlier isolates. The most parsimonious description for VNIc is polyphyletic, with eight different mono- or paraphyletic groups. The MRCA of all VNIc I in the mtDNA tree is the MRCA of 648 isolates, only 89 of which are VNIc.

The most striking incongruity between the mtSNP and the whole-genome data was in the placement of VNIa-5. In the whole-genome tree, VNIa-5 is within the VNIa group with VNIa-4 as its sister taxa. In contrast, in the mtSNP tree, VNIa-5 is an outgroup, even in relation to VNIb and VNIc. There was a 28 bp sequence, intergenic between CNAG_09008 and CNAG_09009 (positions 19,441 to 19,469 of the mtSNP sequence, NC_018792.1), which contained eight variants, present in every VNIa-5 in the dataset. This sequence begins 280 bp downstream of the 3' end of CNAG_09008 and terminates 200 bp upstream of CNAG_09009. It had a per-site substitution rate of 0.28 compared with 0.004 for the VNIa-5 mitochondrial sequence as a whole. None of the variant positions were shared by any other *C. neoformans* strain, or by *C. deneoformans* JEC21 (GCA_000091045) or *C. gattii* R265 (GCA_000149475). When the putative recombinant region was compared against the full nr/nt BLAST database, the closest hit was to *C. neoformans* H99, chromosome 5 (NC_026749.1), positions 80,207 to 80,234, which had 1 bp difference (*E* value = 0.004). This closest sequence on chromosome 5 is within CNAG_06848, which is widely conserved in the fungal kingdom. CNAG_06848 is a 222 bp gene encoding an 'ATP synthase subunit 9, mitochondrial'. There were no strains in our dataset with SNPs in CNAG_06848, which could indicate a reciprocal recombination event. The assembly of the PacBio-sequenced VNIa-5 genome also showed the presence of the highly variable region in the mitochondrial genome

## Discussion

We sequenced 699 isolates of *C. neoformans* covering 19 years and 5 countries on 2 continents, with most isolates derived from two large clinical trials. We integrated our novel data with previously published data[8] to provide extra context for our original findings. This context allowed us to assign 99.4% of the *C. neoformans* isolates sequenced as part of this study to the global clade VNI[3–5]. According to the nomenclature established by Desjardins et al.[8] 98.5% of our isolates belonged to VNIa, compared with 30% of clinical VNI isolates and 18.5% of all isolates sequenced by Desjardins. To some extent, this difference is to be expected due to the focus of Desjardins et al.[8] on both VNI and VNB, and their intensive sampling of Botswana, a known outlier in terms of *Cryptococcus* diversity[3]. This dominance of VNIa in our samples is interesting for two reasons. It raises the question of whether there are specific biological properties of VNIa, or of VNIa-4, VNIa-5 and VNIa-93, which underlie their predominance in our clinical isolates. Secondly, the *C. neoformans* reference strain, H99, belongs to the VNIb lineage, which accounts for fewer than 1.5% of the clinical isolates in our study. We suggest that it may be useful to the *Cryptococcus* research community to consider including more representative isolates (i.e. from VNIa) in detailed laboratory investigations.

The structure of the phylogeny, with large 'flat' clades separated by deep branches, and short internal branches combined with long terminal branches within those clades, is consistent with recent, exponential population expansion[22,23]. That *C. neoformans* should have undergone a recent population expansion is not surprising, considering the increase in the number of susceptible hosts due to the HIV/AIDS epidemic. What is surprising in our dataset is that this rapid expansion is due almost entirely to three sub-clades, with 89% of *C. neoformans* sequenced in this study belonging to the VNIa-4, VNIa-5 and VNIa-93 sub-clades. Two other features consistent with an exponentially increasing population size are (i) the lack of a molecular clock in these sub-clades and (ii) the higher d$N$/d$S$ of SNPs on terminal branches compared with the lineage defining branches. The dominance of these three sub-clades also explains the fact that

even though 97.9% of our *C. neoformans* isolates were VNIa, we observed little diversity within VNIa that was not also observed in the 59 VNIa isolates sequenced by Desjardins et al.[8].

Since the human host is thought to be a dead end for *C. neoformans*, the dominance of these sub-clades cannot be due to amplification through rounds of human infection and release into the environment. Therefore, it either reflects the environmental prevalence in the areas from which our study population become infected, or there could be some properties of these sub-clades which increase their ability to cause human infection. Further ecological and biological studies are required to determine which of these is the case. We also believe that any future environmental sampling work should consider the possible role of a quiescent phase of the *C. neoformans* life cycle, whether that is a spore, desiccated yeast, or the recently described viable but non-culturable form[24]. A resilient quiescent phase would increase the ability of *C. neoformans* to travel long distances, a phenomenon we have observed frequently in our data. Polytomies, which we observed frequently in the *C. neoformans* phylogeny, have also been observed in the phylogenies of bacterial spore formers[25,26].

The phylo-geography of VNIa is characterized by each lineage being predominantly but not exclusively found in a single country or continent. While our sampling is exclusively from Asia and Africa, and is therefore not globally representative, VNIa-4 and VNIa-5 were predominantly Asian (97 and 89%), and VNIa-93 was predominantly African (64%). This finding is consistent with previous reports, with particular STs having been reported to be more common in certain countries, regions or continents[3–5,16]. However, whole-genome sequence (WGS) provides us with extra resolution in resolving whether, for example, the 7% of VNIa-5 strains in Africa are the result of a single introduction or multiple discrete introductions. To address this question, we generated within sub-clade reference genomes using PacBio sequencing and performed within sub-clade phylogenetic analyses. Examination of the within sub-clade phylogenetic trees (Fig. 2) and parsimony analysis shows that international and intercontinental transmission is a frequent event, with 8–13% of internal branches representing an international transmission.

While nearly clonal isolates have been identified in disparate locations by a recent study[9], the authors focussed more on exploring ancient migrations. Our data dramatically illustrate the extent of this on-going intercontinental migration and we offer two alternative explanations. The first potential explanation is that transmission between countries or continents occurs during latent infection, that is, a patient is exposed in one country, and then travels to another country where they develop illness and are sampled. Such long distance latent transmission has been hypothesized previously[27]. Unfortunately, we do not have extensive travel/residence histories for our patients and thus cannot directly address this hypothesis. However, we judge it an unlikely explanation of our findings given the limited extent of contemporary migration between, for exampe, Vietnam and Uganda and the demographics of our patient population (little disposable income). A second, broad hypothesis to explain the large number of transmission events is that they are mediated by environmental factors, either 'natural' or human influenced. Potential natural environmental factors would include air currents or migratory birds; pigeons specifically are considered the most probable vector for global dissemination[28]. Human activities that link the environments of East/Southeast Africa and Southeast Asia include trade in lumber, rice, exotic animals and illegal animal products such as those used in traditional medicine, for exampe, ivory (http://www.aljazeera.com/news/2016/11/exclusive-vietnam-double-standards-ivory-trade-161114152646053.html). While we cannot directly address this hypothesis with our data, airborne spread is well established as a long distance dispersal mechanism

for plant pathogens[29]. Intuitively it might seem unlikely that long-distance airborne dispersal of fungal pathogens occurs frequently. However, if airborne spore dispersal conforms to a non-exponentially bound (or 'fat-tailed') distribution model rather than an exponential model, long-distance dispersions will occur relatively frequently[29,30]. Weather patterns are a proto-typical example of such 'fat-tailed', 'chaotic' (small differences in initial conditions, leading to large differences in outcome) distributions[31]. However, effective quantification of the potential contribution of airborne dispersal is complex[32] and beyond the scope of this paper. Overall, we consider environmental factors to be the better explanation because (i) *Cryptococcus* is fundamentally an environmental organism, (ii) there is limited contemporary human migration between Southeast Asia and East/Southeast Africa and (iii) long-distance dispersal by environmental factors, including wind, is well established for fungal pathogens.

Desjardins et al.[8] established that there is still recombination on-going within VNIa. However, within each sub-clade, recombination appears to be a relatively minor contributor of genetic diversity. LD decay over genomic distance was minimal in all three sub-clades, although the small number of SNPs with a minor allele frequency >0.1 (due to short internal branches) means that this analysis had limited power. However, further evidence against the role of recombination in the generation of within sub-clade diversity is the low proportion of terminal branch SNPs that are homoplasies.

We observed two associations between lineage and clinical phenotype. First, the previously described association between VNIa-5 and the infection of apparently immunocompetent HIV-uninfected patients[15,19], and second, the novel finding of a significantly lower risk of death at 10 weeks in patients infected with VNIa-93, in contrast to previous findings[18]. As VNIa-93 is primarily found in Africa, and outcomes are typically worse in Africa[21], it is notable that we still observed this effect. We also investigated whether there is evidence of within-host evolution reflecting pressure from the host, as has been previously observed[7], by looking for convergent evolution on terminal branches. This analysis found little evidence of significant within-host evolution; there were no known virulence-associated genes with extremes of d$N$/d$S$ due to terminal branch mutations, and only a small proportion (1.5%, 1.8%, 0.4%) of terminal branch SNPs for each sub-clade were homoplasies, and the majority of these were in intergenic regions.

One interesting difference between the VNIa-5 isolates and the rest of VNIa was identified in the mitochondrial sequence. We observed a 21 bp sequence, representing a probable recombination event, which introduced eight SNPs present in every VNIa-5 isolate and absent in every non-VNIa-5 isolate. The most likely candidate for the donor sequence was chromosome 5 of the *C. neoformans* nuclear genome, which encodes a sequence that varies by only 1 bp from the 21 bp putative recombinant fragment. While this has not been described previously in the literature, since the SNPs at these positions were not mixed, and the reads containing the divergent sequence mapped well to the mtDNA, and the PacBio sequence showed the same sequence, we deem it likely that the mitochondrial sequence has been accurately re-constructed, while the origin of the divergent sequence is less certain. That this change occurs in the mitochondrial genome is particularly intriguing as changes in mitochondrial morphology have been reported as underlying the hyper-virulence of the Vancouver outbreak *C. gattii*[33,34]. Whether the changes in the *C. neoformans* mitochondrial genome lead to morphological and in vitro phenotype changes as has been shown for *C. gattii* is an important future research question. The putative recombination is in an intergenic region of the mitochondrial genome, so if this variant underlies a modified phenotype, it is likely driven by changes in gene expression. Fungal mitochondrial

5′ untranslated leader sequences have been described between 81 to 220 bp in length[35], while the putative recombination occurs 200 bp upstream of CNAG_09009.

In summary, our analysis of 699 *Cryptococcus* genomes has revealed that clinical isolates of *C. neoformans* from Vietnam, Laos, Thailand, Uganda and Malawi are concentrated in three main sub-clades. The phylogenetic structure indicates that there has been a recent exponential population expansion of *C. neoformans*, likely due to the increase in the number of people susceptible to infection. Our data show that, unexpectedly, three sub-clades of *C. neoformans* (VNIa-4, VNIa-5 and VNIa-93) have driven this population expansion; the reasons for this remain uncertain and are a key question for future study. Another research question raised by our findings is whether the mitochondrial recombination we observed in VNIa-5 is associated with mitochondrial morphology changes, which could explain the ability of this sub-type to infect HIV-uninfected people. We also show that infection with VNIa-93, which has previously been associated with poorer outcomes, is associated with a significantly reduced risk of death by 10 weeks compared with VNIa-4. Genome sequencing for fungal pathogens can provide insight into clinical and epidemiological features, and pose important future research questions for the field.

## Methods

**Strain selection**. The Vietnamese isolates ($N = 441$) were clinical isolates from the cerebrospinal fluid of patients enroled in a prospective, descriptive study of HIV-uninfected patients with central nervous system infections ($n = 67$) between 1997 and 2014, a randomized controlled trial of antifungal therapy in HIV-infected patients between 2004 and 2011 (http://www.isrctn.com/ISRCTN95123928), the CryptoDex trial (http://www.isrctn.com/ISRCTN59144167) and three environmental isolates from Ho Chi Minh City, Vietnam[11,19–21]. The WGS of eight Vietnamese strains in this analysis have been previously reported[15]. Lao isolates were from 73 patients with invasive cryptococcal infection admitted to Mahosot Hospital, Vientiane, between 2003 and 2015, including 5 from the CryptoDex trial. Isolates from Uganda (132), Malawi (13) and Thailand (40) were all from HIV-infected patients enroled into the CryptoDex trial[21]. Sixty-nine isolates from Vietnam and eight from Laos were derived from patients who were HIV uninfected. All clinical trials had ethical approval from the local IRB in each centre and from the Oxford Tropical Ethics Committee, UK. All participants in clinical trials gave written informed consent.

**Micro and molecular biology**. Isolates were revived from storage by incubation on Sabouraud's agar at 30 °C for 72 h. Single colonies were spread for confluent growth and incubated at 30 °C for 24 h. For Illumina sequencing, genomic DNA was extracted from approximately 0.5 g (wet weight) of yeast cells using the MasterPure Yeast DNA purification kit (Epicentre, USA) according to the manufacturer's instructions. Colonies were grown overnight on YPD media (Merck, UK), suspended in lysis solution, vortexed and incubated at 65 °C for 1 h. Forty micrograms of RNase were added with incubation for 30 min at 37 °C, followed by 70 °C for 10 min. Following cooling on ice, 150 µL of MPC protein precipitation reagent was added with further vortexing (10 s). Cellular debris was pelleted by centrifugation for 10 min at 10,000 rpm. The supernatant was transferred to a clean tube and 500 µL of isopropanol was added, mixed by inversion and re-centrifuged for 10 min. The supernatant was discarded and the pellet was washed with 300 µL of 75% ethanol. Following further centrifugation for 2 min, the ethanol supernatant was discarded and the tube was placed on a 65 °C heat block to dry (30 s to 1 min) DNA and then was re-suspended in 50 µL of dH$_2$O. Whole-genome sequencing was carried out on the Illumina HiSeq 2000 at the Sanger Institute UK, and commercially through Macrogen, Korea using the HiSeq 4000 platform.

DNA for PacBio sequencing was extracted using a modification of the method described at dx.doi.org/10.17504/protocols.io.ewtbfen. High-quality DNA was extracted by bead beating freeze-dried cell pellets, followed by phenol/chloroform purification and DNA precipitation with isopropanol. Briefly, purified single colonies were grown overnight in 5 mL YPD. Cells were washed in phosphate-buffered saline (PBS), suspended in 1 mL PBS and then lyophilized for 3 h using a FIRSTEK BFD 4.5/50 freeze-dryer (FIRSTEK, Middlesex, UK). Lyophilized cells were then homogenized using 3 mm Pyrex beads and a bead beater ($3 \times 60$ s). Homogenized cells were lysed in lysis buffer with 1% polyvinylpyrrolidone at 64 °C prior to the addition of RNAse T1 and proteinase K and resuspension in lysis buffer. Following incubation at room temperature for 1 h with regular tube inversion, 5 M potassium acetate was added with incubation on ice for 5 min. The suspension was spun at $5000 \times g$ at 4 °C for 10 min and repeated. The supernatant was transferred for chloroform:isoamylalcohol (24:1) extraction and again spun at

4 °C, $4000 \times g$ for 10 min. The resultant supernatant was aliquoted and treated with RNase A/T1 and incubated at room temperature for 30 min. Then, the supernatant was aliquoted and mixed with sodium acetate; DNA was precipitated with isopropanol and harvested using pipette tips. PacBio sequencing was performed by Macrogen, Seoul, Korea, for 20 kb SMRT library production, with two SMRT cells per sample, according to the manufacturer's instructions.

**Species identification, principal components analysis**. Species identification was carried out using mash screen function[36] comparing the sample FASTQs against the whole refseq database. For the principal components analysis, all variant positions were loaded into an adegenet[37] (devel branch, commit 43b4360) genlight object using RStudio. Then, the ade4 dudi.pca function was used to determine the principal components. K-means clustering was run on the first two principal components, with values to K between 2 and 10. The total within-cluster sum of squares was plotted for each K, and the number of clusters determined as the 'elbow' in the plot of K vs. total within-cluster sum of squares. As the previously described VNIb and VNIc were grouped into one cluster in the analysis of the first two PCs, the same analysis was carried out on the third and fourth PCs, which separated these two established lineages.

**Phylogenetic analysis**. FASTQ data were mapped against the H99 reference (GCF_000149245) using bwa mem[38], SNPs were called using GATK v3.3.0[39] in unified genotyper mode. Positions where the majority of allele accounted for <90% of reads mapped at that position, which had a genotype quality of <30, coverage <5×, or mapping quality <30 were recorded as *N*s in further analyses. These steps were carried out using the PHEnix pipeline (https://github.com/phe-bioinformatics/PHEnix) and SnapperDB[40]. Positions in which at least one strain had an SNP passing quality thresholds were extracted and used as the input for IQ-TREE v1.6[41] maximum likelihood phylogenetic analysis with the best-fitting model selected by IQ-TREE. Ancestral state reconstruction was carried out using IQ-TREE v1.6[42]. To place our data into the broadest possible context, we included WGS data from Desjardins et al.[8]. To ensure efficient use of computational resources, a preliminary phylogenetic analysis was carried out, including all our data and representatives of VNI, VNBI, VNBII and VNII from Desjardins et al.[8]. For polytomy analysis, ete3[43] was used to delete/collapse nodes (branches) in the tree that contained 0 SNPs. Any node in this new tree with collapsed branches with three or more children was defined as a polytomy. Pacbio data was assembled using Canu v1.5[44] and default parameters, polishing with Illumina data from the corresponding isolate using Pilon v1.22[45] for multiple rounds until the number of indels being corrected per round was <2. We searched for nonsense mutations in the following 34 genes (CNAG_00178, CNAG_00550, CNAG_00572, CNAG_00612, CNAG_00720, CNAG_00770, CNAG_00772, CNAG_01037, CNAG_01642, CNAG_01916, CNAG_02073, CNAG_02490, CNAG_03449, CNAG_05201, CNAG_05862, CNAG_06724, CNAG_07552, CNAG_07599, CNAG_00299, CNAG_00328, CNAG_01163, CNAG_02512, CNAG_02771, CNAG_03160, CNAG_04733, CNAG_05102, CNAG_05531, CNAG_06384, CNAG_02467, CNAG_02544, CNAG_05198, CNAG_05537, CNAG_05746, CNAG_06143), which includes the ERCC, MLH, MSH and RAD families of proteins.

**Analysis of effect of sub-clade on outcome**. We assessed the effect of sub-clade on time to death (10 weeks and 6 months) in HIV-infected patients with cryptococcal meningitis with a Cox proportional hazards regression model with sub-clade as the main covariate. We included all patients with available data from our two randomized controlled trials. The model was adjusted for country, induction antifungal treatment (amphotericin monotherapy for 4 weeks, amphotericin combined with flucytosine for 2 weeks, or amphotericin combined with fluconazole for 2 weeks) and the use of adjunctive treatment with dexamethasone[20,21]. We tested the proportional hazard assumption based on scaled Schoenfeld residuals. Since we knew from the Cryptodex trial that the covariate dexamethasone does not satisfy this assumption, we included a time varying coefficient for dexamethasone use.

**Recombination analysis**. Recombination analysis was carried out independently for VNIa-4, VNIa-5 and VNIa-93. LD ($R^2$) was calculated on a per-lineage basis using vcftools v0.1.14[46] and the –geno-r2 option and a minimum allele frequency (MAF) of 0.1, LD was grouped in 100,000 bp windows as there were not many SNPs with an MAF >0.1 within sub-clades due to the short internal branches.

**Reporting summary**. Further information on research design is available in the Nature Research Reporting Summary linked to this article.

## Data availability

All sequence data from this study is publicly available at EBI ENA under BioProject accessions PRJEB27222 and PRJEB5282. Per-sample accessions are available in Supplementary Data 1, which is available as an Excel file from https://doi.org/10.6084/m9.figshare.7796147.v1. All other data are available upon request.

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

## Acknowledgements

This work was supported by a Wellcome Trust Intermediate fellowship WT097147MA to J.N.D. and by the United Kingdom Department for International Development, the Wellcome Trust and the Medical Research Council through a grant (G1100684/1) from the Joint Global Health Trials program, which is part of the European and Developing Countries Clinical Trials Partnership, supported by the European Union. We would like to acknowledge the contribution of the Pathogen Informatics team at the Wellcome Sanger Institute, the Sequencing team at the Wellcome Sanger Institute, Macrogen of South Korea and MRC CLIMB for providing computational capacity[47]. Isolates from Laos were obtained as part of the work program of the Lao-Oxford-Mahosot Hospital Wellcome Trust Research Unit funded by the Wellcome Trust (106698/Z/14/Z). We are grateful to all the laboratory and clinical staff who helped with the collection of the isolates and data.

## Author contributions

Designed the study: J.N.D., P.M.A., S.B., S.H. Provided samples or data: J.N.D., J.B., F.K., W. C., D.A.B.D., S.R., V.D., L.Q.H., N.V.V.C., N.L.N.T., A.K.C., G.E.T., D.G.L., G.D. Performed experiments and lab work: L.T.T., P.H.T., D.V.A., N.M.T. Performed analyses: P.M.A., L.T. H.N., J.N.D. Wrote paper: P.M.A., C.A., J.N.D. All authors reviewed the final draft.

## Additional information

**Competing interests:** The authors declare no competing interests.

