## [Peer Review File · Nature Communications]

Reviewers' comments:

Reviewer #1 (Remarks to the Author):

This is a well-written paper on a very large panel of sequenced genomes of the main globalised lineage of *Cryptococcus neoformans*, VNI. This is a major sequencing effort that has been comprehensively analysed, and has led to some well-argued insights into the biology of VNI.

The main limitation of the paper is the lack on any experimental test of patterns of virulence seen in patients translated to a suitable animal model, with accompanying genetic manipulations to confirm molecular hypotheses. That said, this is a substantial piece of work that provides a welcome extension of the two main previous global *C. neoformans* papers by Desjardins and Rhodes.

- 336 "while a recently increased growth rate could be due to exploitation of a new niche, such as the HIV infected human host". This seems to be experimentally tractable. Given that a main finding (and argument) in the paper is that the long terminal branches result from bursts of growth (potentially within an HIV patient as suggested here), that can be tested by allowing a burst of growth the occur in order to directly measure mutation accumulation (there is something along this line described in Rhodes et al in G3, but that data is not used, or commented on, here). The Rhodes et al paper also documents hypermutators that accumulate mutations owing to defects in mismatch repair. As this could be a mechanism that accounts for some of the (longer) branches in the dataset, then checking for nonsense mutations in the mismatch repair genes seems an obvious point of enquiry.

- 359. Historically, there has been alot of migration between Asia and Africa - just look at the amount of Asians in Africa. For instance, the Tsavo railway was built by Asian 'coolies' so there were likely substantial trafficking during the British Colonial era.

Reviewer #2 (Remarks to the Author):

The manuscript by Ashton et al describes genomic analysis of a large collection of *Cryptococcus neoformans* from the South-East Asia and Africa. Authors describe extensive clonality among isolates, as highly genetically related isolates are found on different continents. They also show the apparent absence of the molecular clock-like relationships among closely related strains and the extensive polytomy of terminal branches on phylogenetic trees, which they interpret as an indication of the presence of the quiescent state in *Cryptococcus* life cycle, which prevents it from actively growing/mutating in the environment. Finally, they show associations between genotypes and clinically relevant phenotypes by demonstrating the reduced mortality of patients infected with VNIa-93 genotype and confirming the previously shown affinity of VNIa-5 (ST5) for immunocompetent patients.

The genomic analysis of such large collection of mostly Asian *C. neoformans* is certainly of interest, although the approach is not particularly new, as similar studies were published by Rhodes et al. (Genet 2017) and Desjardins et al (Gen Res, 2017). For the most parts, the analysis is well done and the results are scientifically sound. It would be helpful to show bootstrap values for the trees, which I appreciate is technically challenging but has been done by others using WGS data.

However, I strongly disagree with the authors' interpretation of the results. There is absolutely no indication of the presence of the quiescent state in *Cryptococcus* life cycle and the following provides the evidence against it:

- *Cryptococcus* is widespread in the environment and can be easily isolated in culture from the environmental sources, which is not consistent with the quiescent state. Aged pigeon dropping are often "packed" with *Cryptococcus* cells, which is often the main yeast isolated from these samples.

- Both *C. neoformans* and *C. gattii* can readily grow on culture media that consist of the autoclaved pigeon droppings (Nielsen et al, EC 2007) and boiled tree bark (Litvintseva et al, PLoS One, 2011) as sole nutrient sources indicating that there is no reason for *Cryptococcus* not to grow on these substrates in the environment.
- There is no apparent genetic difference between environmental and clinical isolates of the same genotype: following the authors' logic, environmental isolates should reside on the shorter terminal branches compared to the clinical ones, which definitely was not evident in Desjardins et al study that included environmental strains.
- Finally, there is no evidence for the accelerated mutation rate either inside the host or in culture. Chen et al (mBio 2017) identified 0-6 SNPs in serially collected isolates from patients over 6 months. Arras et al (Sci Rep, 2017) identified 32 SNPs accumulated between 1978 and 2017 in the descendants of the original H99 strain isolated from a patient in 1978, which corresponds to less than 1 SNP per year and is inconsistent with the authors' hypothesis of the accelerated growth/mutation upon isolation.

Instead, a much simpler and more parsimonious explanation for the observed polytomy and the lack of molecular clock is the demographic history, which in *Cryptococcus* was shaped by the two processes: first, severe bottle-neck/emergence of a small number of strains from the ancestral population, followed by the second, rapid expansion in the new habitat. In fact, polytomy on a tree is a classic example of population expansion, which often happens in response to the new habitat. The apparent lack of molecular clock is a consequence of the very large effective population size which by itself is a consequence of the rapid global expansion.

Reviewer #1 (Remarks to the Author):

This is a well-written paper on a very large panel of sequenced genomes of the main globalised lineage of *Cryptococcus neoformans*, VNI. This is a major sequencing effort that has been comprehensively analysed, and has led to some well-argued insights into the biology of VNI.

The main limitation of the paper is the lack on any experimental test of patterns of virulence seen in patients translated to a suitable animal model, with accompanying genetic manipulations to confirm molecular hypotheses. That said, this is a substantial piece of work that provides a welcome extension of the two main previous global *C. neoformans* papers by Desjardins and Rhodes.

- 336 "while a recently increased growth rate could be due to exploitation of a new niche, such as the HIV infected human host". This seems to be experimentally tractable. Given that a main finding (and argument) in the paper is that the long terminal branches result from bursts of growth (potentially within an HIV patient as suggested here), that can be tested by allowing a burst of growth to occur in order to directly measure mutation accumulation (there is something along this line described in Rhodes et al in G3, but that data is not used, or commented on, here). The Rhodes et al paper also documents hypermutators that accumulate mutations owing to defects in mismatch repair. As this could be a mechanism that accounts for some of the (longer) branches in the dataset, then checking for nonsense mutations in the mismatch repair genes seems an obvious point of enquiry.

We apologise for the lack of clarity in this section. The intended message of the quoted sentence was that the identification of large, genetically closely related clades (i.e. VNIa-4, 5 and 93) could be due to a recent clonal expansion of the organism into a new niche, the HIV infected human host. We have added some text to L435-437 which clarifies this.

While it is likely that some of the mutations on the terminal branches occur within patients, our finding of long terminal branches in environmentally isolated strains indicates that long branches are not exclusively due to within patient growth. Clarification on this point has been added to L522-523.

In our opinion, the best approach to investigate mutation accumulation within host is that of e.g. Chen et al. (2017), looking at paired isolates from the same patients separated by some time. Therefore, we decided to extrapolate from this data. This has been expanded on in lines 559-563. Rhodes et al. 2017 in G3 don't describe in vitro mutation accumulation experiments, but do report on mutation rates inferred from the differences between initial infection and recrudescence. They report a very high mutation rate of 3 SNPs per day for VNI, and even higher rates of 365 and 12 SNPs per day in VNB and VNII. We have included a brief discussion of their findings on L563-567.

The reviewer is also correct to suggest that investigating mutations in DNA mismatch repair genes would be interesting. We have added the methods of our investigation of these on L809-816 and the results on L297-305. Unfortunately, nothing of note was found.

- 359. Historically, there has been a lot of migration between Asia and Africa - just look at the amount of Asians in Africa. For instance, the Tsavo railway was built by Asian 'coolies' so there were likely substantial trafficking during the British Colonial era.

The reviewer is correct to point out the amount of historical population movement between Asia and Africa. What we meant in this section is that we don't think it's likely that our Vietnamese/Asian patient population has travelled back and forth to Africa, transmitting *Cryptococcus* between the countries through human infection. We think it's more likely to be the result of either 'natural' or 'man-made' environmental transmission e.g. wind or lumber trade. We have changed the text to clarify (L474-476).

Reviewer #2 (Remarks to the Author):

The manuscript by Ashton et al describes genomic analysis of a large collection of *Cryptococcus neoformans* from the South-East Asia and Africa. Authors describe extensive clonality among isolates, as highly genetically related isolates are found on different continents. They also show the apparent absence of the molecular clock-like relationships among closely related strains and the extensive polytomy of terminal branches on phylogenetic trees, which they interpret as an indication of the presence of the quiescent state in *Cryptococcus* life cycle, which prevents it from actively growing/mutating in the environment. Finally, they show associations between genotypes and clinically relevant phenotypes by demonstrating the reduced mortality of patients infected with VN1a-93 genotype and confirming the previously shown affinity of VN1a-5 (ST5) for immunocompetent patients.

The genomic analysis of such large collection of mostly Asian *C. neoformans* is certainly of interest, although the approach is not particularly new, as similar studies were published by Rhodes et al. (Genet 2017) and Desjardins et al (Gen Res, 2017). For the most parts, the analysis is well done and the results are scientifically sound. It would be helpful to show bootstrap values for the trees, which I appreciate is technically challenging but has been done by others using WGS data.

We have added bootstrap values to Figures 1 and 2. The very short branches are poorly supported, as expected, but this has no impact on our interpretation. The vast majority of longer branches are well supported.

However, I strongly disagree with the authors' interpretation of the results. There is absolutely no indication of the presence of the quiescent state in *Cryptococcus* life cycle and the following provides the evidence against it:

Two potential candidates for the quiescent state of the *Cryptococcus* lifestyle are the basidiospore and desiccated yeast forms, and we believe that these are accepted to be

relevant to the *Cryptococcus* life cycle and ecology (Heitman & Lin, 2006, PMID 16704346; Velagapudi et al., 2009, PMID 19620339; Botts et al., 2009, PMID 19181873).

- *Cryptococcus* is widespread in the environment and can be easily isolated in culture from the environmental sources, which is not consistent with the quiescent state. Aged pigeon dropping are often “packed” with *Cryptococcus* cells, which is often the main yeast isolated from these samples.

The reviewer is correct to point out that there are well known reservoirs of *Cryptococcus* in the environment (pigeon guano). Our thinking on this matter was informed by extensive environmental sampling conducted by our laboratory (>5000 samples, >79 tree species) which resulted in the isolation of 120 isolates of *Cryptococcus* species organisms, but only 3 *C. neoformans* organisms. We did not include this data in our initial submission of this manuscript as a separate manuscript was planned. However, after discussions with the lab members involved, we have included this data here. The methods are on lines 699-780, results on L145-147 and discussion on L421-430.

- Both *C. neoformans* and *C. gattii* can readily grow on culture media that consist of the autoclaved pigeon droppings (Nielsen et al, EC 2007) and boiled tree bark (Litvintseva et al, PLoS One, 2011) as sole nutrient sources indicating that there is no reason for *Cryptococcus* not to grow on these substrates in the environment.

We sampled 1000 tree trunks from more than 79 species during our environmental sampling, isolating only 3 *C. neoformans* organisms, although we isolated >120 other *Cryptococcus* species.

- There is no apparent genetic difference between environmental and clinical isolates of the same genotype: following the authors’ logic, environmental isolates should reside on the shorter terminal branches compared to the clinical ones, which definitely was not evident in Desjardins et al study that included environmental strains.

It would have made an interesting story if the environmental isolates had short terminal branches, indicating that significant growth only occurred in patients. However this was not the case and our interpretation does not require this. Having observed long terminal branches in the environmental organisms as well, our interpretation is that in order to either infect the human host or be cultured from the environment *C. neoformans* must be actively growing. In this way, the cultured organisms are not representative of the typical environmental organism, because we infer from the short internal branches that environmental organisms frequently don’t undergo a large amount of growth. We have added a figure laying out our hypothesis to the discussion section (Figure 4).

- Finally, there is no evidence for the accelerated mutation rate either inside the host or in culture. Chen et al (mBio 2017) identified 0-6 SNPs in serially collected isolates from patients over 6 months. Arras et al (Sci Rep, 2017) identified 32 SNPs accumulated between 1978 and 2017 in the descendants of the original H99 strain isolated from a patient in 1978, which corresponds to less than 1 SNP per year and is inconsistent with the authors’ hypothesis of the accelerated growth/mutation upon isolation.

Our interpretation of our data does not rely on an accelerated mutation rate either inside the host or in laboratory culture and we have not made any claims about accelerated mutation rates. As an imperfect proxy for a molecular clock, we extrapolated from the rate reported by Chen et al., finding that a median terminal branch accounts for 7 to 12 years, which is an unfeasible length of time to have an uncontrolled *C. neoformans* infection.

Instead, a much simpler and more parsimonious explanation for the observed polytomy and the lack of molecular clock is the demographic history, which in *Cryptococcus* was shaped by the two processes: first, severe bottle-neck/emergence of a small number of strains from the ancestral population, followed by the second, rapid expansion in the new habitat. In fact, polytomy on a tree is a classic example of population expansion, which often happens in response to the new habitat. The apparent lack of molecular clock is a consequence of the very large effective population size which by itself is a consequence of the rapid global expansion.

We agree that bottle-neck/emergence into a new niche from a small number of initial strains best explains the overall structure observed in VN1a, i.e. VN1a-4/5/93 being large 'flat' clades separated by long branches. We have modified the relevant section of our discussion to better reflect this (see L435-440). However, we don't think that this same effect is responsible for the polytomies within those sub-clades (VN1a-4/5/93), because there are a large number of separate polytomies, each one representing a different population expansion event. It is difficult to link this with the scenario of the HIV infected human, as humans are thought to be a dead end for *C. neoformans*. We think that each polytomy within VN1a-4/5/93 represents a single population entering the quiescent phase at the same time due to i.e. nutrient limitation. The quiescent particles are then dispersed (sometimes between continents) before finding themselves in a nutrient rich environment in which it is well suited to grow (e.g. bird guano, animal corpse, etc) and undergoing population expansion. We have added this text to L546-561.

Reviewers' comments:

Reviewer #1 (Remarks to the Author):

The reviewed paper does well with what it has at its disposal - a large sample of sequenced *Cryptosporidium* genomes - however this doesn't really help with the main inference, that *C. neoformans*' growth is rare and highly stochastic, leading to 'explosions' of a genotype which undergoes intercontinental dispersal. This is a genomic observation predicated upon the structure of the phylogeny, however is begging experimental testing. This results in the rather weak conclusion of the abstract... 'Based on these data, we hypothesise that *C. neoformans* VN1a spends much of its time in the environment in a quiescent state, while, when it is sampled, it has almost always undergone an extended period of growth.' This sentence is rather equivocal - what does 'sampled' mean here to the reader? Infection I take it but that's not clear. From L423 'but the more likely explanation is that *C. neoformans* VN1a has a recently increased growth rate due to exploitation of a new niche, such as the HIV infected human host.' suggests that the accumulation of mutation is predicated upon growth in the human. However we know that the human is a dead end host, so this must therefore be based on substantial accumulation of diversity in the hosts CNS (such as proposed by Rhodes & Chen, albeit with different measurements of diversity accumulation). Needs clarification.

Really, more work needs to be done on the life history of *Cryptosporidium* however I realise this is outside the reach of this study. Perhaps therein lies the real value of this study - to stimulate that research.

Reviewer #2 (Remarks to the Author):

I appreciate the large amount of work that went into this manuscript and the authors' efforts to address my previous questions. However, based on the reasons below, I still do not agree with one of the main conclusions about the existence of a quiescent state in *C. neoformans*:

A. Authors' main argument is that phylogenetic structure with long terminal and short internal branches combined with the absence of molecular clock indicative is of a quiescent state. As mentioned previously, this structure represents a classical signature of an expanding population after a bottleneck event. In the attached file, there is an excerpt from a textbook (The phylogenetic handbook: a practical approach to phylogenetic analysis and hypothesis testing. Salemi, M., Vandamme, A. M., & Lemey, P. (Eds.). (2009). Cambridge University Press) that describes exactly this type of phylogenetic structure, which is not uncommon in fungal pathogens.

B. The authors' second argument is based on their inability to culture *C. neoformans* from the environment, and I can think of at least two reasons why their environmental sampling did not work:

1. Although intellectually appealing, randomized environmental sampling guided by the grid almost never works for isolation of human pathogenic fungi from the environment. This approach has been tried by several groups for isolation of *Cryptococcus*, *Coccidioides* and *Histoplasma*, and to my knowledge, has never been successful. These fungi have "spotty" distribution and are adapted to narrow specialized niches, such as aged bird guano, abandoned animal borrows, wood hollows inhabited by bats, etc. Such microhabitats are missed by random sampling guided by the grid.
2. I am surprised by the authors' focus on sampling arboreal sites. Although *C. gattii* and *C. neoformans* VNB strains are frequently isolated from trees and surrounding soil, the affinity of VNI for aged pigeon guano is well documented (<https://www.ncbi.nlm.nih.gov/pubmed/21589919>; <https://jcm.asm.org/content/43/2/556.long>). This is especially true for strains with globally distributed genotypes, such as VN1a-4, VN1a-5, VN1a-93 and others, for whom pigeons have been

implicated in the global expansion. Out of staggering 5,000 tested samples, only 200 were avian dropping, and it remains unclear how many of those were from pigeons and were sufficiently "aged" and protected from UV to support *Cryptococcus* growth.

There are some additional points as well:

- Isolation of other species of *Cryptococcus* does not necessarily mean that the habitat is suitable for *C. neoformans*: different species may have different requirements, other species of *Cryptococcus* are rarely co-isolated with *C. neoformans*.
- There is no evidence of a quiescent state in the *Cryptococcus* life cycle: both basidiospores and desiccated yeast cells readily germinate and propagate under suitable conditions. To my knowledge, there are no similar examples in other fungi as well.
- In addition, the disease cryptococcosis is highly prevalent in different parts of the world including Vietnam, I have had a hard time reconciling how *Cryptococcus* could be quiescent but widespread at the same time: in order to colonize large geographic areas the organism needs to propagate in the environment.

Ana Litvintseva

Authors argue that the phylogenetic structure with long terminal and short internal branches combined with the absence of molecular clock is suggestive of a quiescent state. As mentioned previously, this structure represents a classical signature of an expanding population after a bottleneck event. Below an excerpt from a textbook that describes exactly this type of phylogenetic structure, which is apparently not uncommon in fungal pathogens.

[Redacted]

The phylogenetic handbook: a practical approach to phylogenetic analysis and hypothesis testing. (Salemi, M., Vandamme, A. M., & Lemey, P. (Eds.). (2009). Cambridge University Press.

a.

Reviewers' comments:

Reviewer #1 (Remarks to the Author):

The reviewed paper does well with what it has at its disposal - a large sample of sequenced Crypto genomes - however this doesn't really help with the main inference, that *C. neoformans*' growth is rare and highly stochastic, leading to 'explosions' of a genotype which undergoes intercontinental dispersal. This is a genomic observation predicated upon the structure of the phylogeny, however is begging experimental testing. This results in the rather weak conclusion of the abstract...'Based on these data, we hypothesise that *C. neoformans* VN1a spends much of its time in the environment in a quiescent state, while, when it is sampled, it has almost always undergone an extended period of growth.' This sentence is rather equivocal - what does 'sampled' mean here to the reader? Infection I take it but that's not clear.

In light of the reviews and editor's comments we have re-interpreted our data and have removed reference to quiescence, although we were interested to see the recent publication on BioRxiv from Alexandre Alanio's group regarding the existence of VNBC – viable but non-culturable forms of *Cryptococcus* (link here). We think that the exploration of the ecological and epidemiological significance of these biotypes will be an interesting field for future study. As we have re-written the abstract in the context of this re-interpretation, we hope this addresses this point of reviewer 1.

From L423 'but the more likely explanation is that *C. neoformans* VN1a has a recently increased growth rate due to exploitation of a new niche, such as the HIV infected human host.' suggests that the accumulation of mutation is predicated upon growth in the human. However we know that the human is a dead end host, so this must therefore be based on substantial accumulation of diversity in the hosts CNS (such as proposed by Rhodes & Chen, albeit with different measurements of diversity accumulation). Needs clarification.

In light of the re-interpretation of our data, we have re-written this section of the paper.

Really, more work needs to be done on the life history of Crypto however I realise this is outside the reach of this study. Perhaps therein lies the real value of this study - to stimulate that research.

Reviewer #2 (Remarks to the Author):

I appreciate the large amount of work that went into this manuscript and the authors' efforts to address my previous questions. However, based on the reasons below, I still do not agree with one of the main conclusions about the existence of a quiescent state in *C. neoformans*:

A. Authors' main argument is that phylogenetic structure with long terminal and short internal branches combined with the absence of molecular clock indicative is of a quiescent state. As mentioned previously, this structure represents a classical signature of an expanding population after a bottleneck event. In the attached file, there is an excerpt from a textbook (The phylogenetic handbook: a practical approach to phylogenetic analysis and hypothesis testing. Salemi, M., Vandamme, A. M., & Lemey, P. (Eds.). (2009). Cambridge University Press) that describes exactly this type of phylogenetic structure, which is not uncommon in fungal pathogens.

Thank you for sharing this reference, it was very helpful in the re-interpretation of our data. We have withdrawn our hypothesis about the importance of the quiescent phase in the natural history of *C. neoformans* infections and instead focussed on what the phylogeny tells us about the exponential growth of *C. neoformans* population. This growth may be expected, due to the increase in the number of susceptible hosts, but the surprising finding in our data is that *C. neoformans* population growth, in our study populations, has been restricted to three clonal sub-clades, VN1a-4, VN1a-5, and VN1a-93. We can only hypothesise about the reason for the success of these sub-clades, but we hope that our findings stimulate further biological and ecological investigations to address this question. The re-interpretation of our results are presented on lines 359-405 of the discussion. We have limited our discussion of the quiescent phase to L315-321.

B. The authors' second argument is based on their inability to culture *C. neoformans* from the environment, and I can think of at least two reasons why their environmental sampling did not work:

We have removed the environmental sampling results from the paper and re-visited the conclusions. Because we have withdrawn these data and the associated arguments we have not addressed the comments below.

1. Although intellectually appealing, randomized environmental sampling guided by the grid almost never works for isolation of human pathogenic

fungi from the environment. This approach has been tried by several groups for isolation of *Cryptococcus*, *Coccidioides* and *Histoplasma*, and to my knowledge, has never been successful. These fungi have “spotty” distribution and are adapted to narrow specialized niches, such as aged bird guano, abandoned animal borrows, wood hollows inhabited by bats, etc. Such microhabitats are missed by random sampling guided by the grid.

2. I am surprised by the authors’ focus on sampling arboreal sites. Although *C. gattii* and *C. neoformans* VNB strains are frequently isolated from trees and surrounding soil, the affinity of VNI for aged pigeon guano is well documented (<https://www.ncbi.nlm.nih.gov/pubmed/21589919>; (<https://jcm.asm.org/content/43/2/556.long>). This is especially true for strains with globally distributed genotypes, such as VN1a-4, VN1a-5, VN1a-93 and others, for whom pigeons have been implicated in the global expansion. Out of staggering 5,000 tested samples, only 200 were avian dropping, and it remains unclear how many of those were from pigeons and were sufficiently “aged” and protected from UV to support *Cryptococcus* growth.

There are some additional points as well:

- Isolation of other species of *Cryptococcus* does not necessarily mean that the habitat is suitable for *C. neoformans*: different species may have different requirements, other species of *Cryptococcus* are rarely co-isolated with *C. neoformans*.
- There is no evidence of a quiescent state in the *Cryptococcus* life cycle: both basidiospores and desiccated yeast cells readily germinate and propagate under suitable conditions. To my knowledge, there are no similar examples in other fungi as well.
- In addition, the disease cryptococcosis is highly prevalent in different parts of the world including Vietnam, I have had a hard time reconciling how *Cryptococcus* could be quiescent but widespread at the same time: in order to colonize large geographic areas the organism needs to propagate in the environment.

Ana Litvintseva

REVIEWERS' COMMENTS:

Reviewer #2 (Remarks to the Author):

I appreciate authors' patience and efforts to modify the manuscript, it looks great now. I have no further comments